# Heritability Estimates of Age at First Calving and Correlation Analysis in Angus Cows Bred in Hungary

**DOI:** 10.3390/ani14243715

**Published:** 2024-12-23

**Authors:** Judit Márton, Szabolcs Albin Bene, Ferenc Szabó

**Affiliations:** 1Hungarian Hereford, Angus and Galloway Breeders Association, Dénesmajor 2, H-7400 Kaposvár, Hungary; 2Georgikon Campus, Hungarian University of Agriculture and Life Sciences, Deák Ferenc U. 16, H-8360 Keszthely, Hungary; 3Albert Kázmér Faculty of Agriculture and Food Sciences, Széchenyi István University, Vár T. 2, H-9200 Mosonmagyaróvár, Hungary

**Keywords:** Angus, age at first calving, population genetic parameters, genetic trend, heritability, breeding value

## Abstract

This paper summarizes the results of a study on the age at first calving (AFC) in Angus cows bred in Hungary. This trait is critical for economic and sustainability reasons due to its relationship with effectiveness and profitability. The results show significant effects of the birth season, herd, and the sire on this trait. No phenotypic or genetic changes were found during the 23-year study period. However, higher heritability estimates than previously reported suggest the possibility of the selection of the AFC in the Angus population.

## 1. Introduction

The profitability and sustainability of beef herds are significantly influenced by the age at first calving (AFC), which has a direct impact on the reproductive and productive performance of the cows. Regarding the economic sustainability of beef cattle farming, reproductive performance is a decisive factor. According to Pulina et al. [1], the main limitations in the beef sector include unfavorable reproduction, lower meat yield per live weight, significantly lower productivity, and a longer production cycle compared to other livestock species. Heifers that calve earlier in the calving period give approximately one more calf during their lifetime than their counterparts that calve later. Thus, the profitability and sustainability of the cow–calf operation are significantly influenced by the cow’s productive life and AFC [2]. The AFC does not depend only on environmental conditions, biological factors, and genetic background; in many cases, it also requires a strategic decision that takes into account economic efficiency and long-term production goals. It is influenced by breeders’ concern that an earlier AFC may increase the risk of calving difficulties, reduce the success of re-pregnancy, and have an adverse effect on the cow’s subsequent performance [3,4].

During a study of Blonde d’Aquitaine cows, López-Parades et al. [5] found that reducing the AFC from 3 to 2 years reduced feed cost by 21.24 USD/year, reduced production cost by 26.52 USD/year, and led to an additional profit of 25.80 USD/year per calf slaughtered over the productive lifespan of the cow. An early AFC (22–24 months) has significant advantages in beef cattle breeding: it can ensure a faster return, increase both the number of calves weaned and meat production and reduce the replacement rate of cows [5]. If AFC was less than or equal to 24 months, 0.7 more calves were weaned by age 6.5 compared to an AFC of 36 months. Early calving heifers achieved a gain of 36.15 USD by the end of their fourth year of life, or 500 USD/cow over the cow’s lifetime [6].

Appropriate feeding and genetic selection are critical factors in optimizing AFC, especially in temperate regions. Based on data from the Irish Cattle Breeding Federation and Teagasc [7], reducing the AFC from 36 months to 24 months results in a smaller (−12%) carbon footprint of 11.2 kg CO_2_ eq/kg live weight (24-month AFC) compared with 12.7 kg CO_2_ eq/kg live weight (36 months AFC); thus, a saving of Euro 114/cow/year can be achieved. Examining data from the United States Department of Agriculture, Moorey and Biase [8] estimate that the calving of 1.6 million heifers in the USA later than 23–27 months of age equates to a loss of 210 million dollars to the beef sector as a result of late breeding.

Maximum production efficiency can be achieved without complications by calving at 2 years of age, with less than 5% calf mortality and a calving interval of 365 days. Based on data from Great Britain, 8.5% of beef heifers calve before the expected age of 24 months [9]. Cushman et al. [10] found that heifers that calved earlier in the breeding season stayed in the herd longer, and their calves were weaned with a higher total weight than their counterparts that calved later. Hickson et al. [4] found that reducing the AFC of heifers from 3 to 2 years of age significantly increases fertility and the number of calves weaned annually from 2-year-old heifers who can be fed without assistance.

There are many literature sources on AFC in different beef cattle breeds. Based on the information listed in Table 1, the h^2^ value of AFC is low to moderate. It can be observed that the value of AFC is typically between 2 and 3 years for breeds belonging to the British maternal line and continental terminal breeds, while it is typically over 3 years for late-maturing indigenous and Zebu-type cattle.

According to Twomey and Cromie [21], the AFC does not affect the performance of mature cows in terms of a cow’s longevity or live weight, calving interval and difficulty, or weaning weight of their calves. The genetic correlation between AFC and second–third calving intervals (+0.40) suggests that selection on earlier AFC can reduce the calving interval [22]. Roughsedge et al. [23] and Berry and Evans [14] reported a negative genetic correlation for these traits. Boligon et al. [24] report negative genetic correlations of −0.29 and −0.24 between AFC and post-weaning weight and yearling weight, respectively. The genetic correlation between AFC and the total number of calvings indicates that heifers who calve earlier produce more calves during their lifetime [5]. There are numerous datasets addressing the relationship between AFC and other reproductive traits, with diverse and often conflicting information.

Precise knowledge of the genetic characteristics of the Angus breed in Hungary is available in only a small number of populations. Therefore, our study aimed to determine the average AFC and its heritability. Moreover, we aimed to examine the effect of various environmental factors on AFC, as well as phenotypic and genetic trends and the breeding value (BV) of Angus sires.

Among the breeding objectives of the Hungarian Hereford, Angus, and Galloway Breeders’ Association (HHAGBA), the improvement of reproductive traits is given a prominent role in the sustainability of beef cattle farming. AFC is of paramount importance, as it has a direct impact on reproductive efficiency and lifetime productivity.

To achieve optimal AFC, we recommend targeted genetic selection, improved herd management practices, and nutritional strategies adapted to Hungarian production systems. Breeders should consider using selection indices that include AFC to improve economic performance and sustainability.

## 2. Materials and Methods

### 2.1. The Database

The pedigree and performance testing data of the HHAGBA were used for this study. Data for the AFC of cows born between 1998 and 2021 were analyzed. The dataset included 2955 pedigree Angus and high (>75%) blood Angus cows (Table 2) in five different genetic groups [25].

The studied population consisted of 2106 Angus dams and 200 purebred Angus sires. Only individuals with known maternal and paternal lineage were included in the analysis. When determining the date of the first calving, only live calves were taken into account (aborted and stillborn calves were not considered), and a total of 199 outliers under 19.0 months (2) and over 46.0 months (179) were excluded. The AFC was determined by the difference between the cow’s date of birth and the date of first calving.

Genetic groups were distinguished by origin, color variant, size, and type: group 1: the large-framed modern type of Canadian and American red Angus; group 2: traditional type of red Angus; group 3: traditional, exclusively British type; group 4: traditional British type black and American imported red Angus; group 5: a mixture of individuals from the other 4 groups.

The normality of the data was tested using the Kolgomorov–Smirnov test, while the homogeneity of the data was assessed by Levene’s test.

The data were selected by the Hungarian Hereford, Angus, and Galloway Breeders’ Association’s own database (HHAGBA registry). Data preparation was carried out using the programs Microsoft Excel and Word 2021. The evaluation and the correlation matrix were conducted using IBM SPSS Statistics for Windows, Version 27.0 [26].

### 2.2. Examining the Effects of Different Factors

The effects of fixed and random factors influencing the trait AFC were evaluated using a general linear model (GLM) within Anova Type III univariate analysis of variance (Table 3) before running the genetic analysis using the BLUP animal model. When compiling this model, sire (sire of the cow) was included as a random effect, while the other factors under examination (herd, color variant of the cow, birth year of the cow, and birth season of the cow) were included as fixed effects. The estimation model was as follows:(1)$${\overset{\wedge }{y}}_{hijlkl}=\mu +S_{h}+F_{i}+C_{j}+Y_{k}+M_{l}+e_{hijkl}$$
where ŷ_hijkl_ = the estimate of the AFC for a cow from sire “h”, in the herd “i”, with “j” color, in “k” birth year, and “l” birth season; μ = the overall mean of all observations; S_h_ = the random effect of the sire; F_i_ = the fixed effect of the herd; C_j_ = the fixed effect of the cow’s color variant; Y_k_ = the fixed effect of the birth year; M_l_ = the fixed effect of the birth season, and e_hijkl_ = random error.

### 2.3. Estimation of Population Genetic Parameters

The available database allowed for both a simpler sire model and a more complex animal model. Population genetic parameters were determined using the GLM [16] and BLUP (Best Linear Unbiased Prediction) [27] animal models.

The population genetic parameters determined by the GLM model were σ^2^_s_ = sire variance, σ^2^_e_ = residual variance, σ^2^_p_ = phenotypic variance, and h^2^_s_ = heritability estimated based on sire variance. This differs from the BLUP model, which directly estimates additive genetic variance.

Heritability value (h^2^) was calculated as additive genetic variance (σ^2^_a_) divided by phenotypic variance (σ^2^_p_) using the following formula [28]:(2)$$h^{2}=\frac{{\sigma^{2}}_{a}}{{\sigma^{2}}_{a}+{\sigma^{2}}_{e}}=\frac{{\sigma^{2}}_{a}}{{\sigma^{2}}_{p}}$$
where h^2^ = heritability; σ^2^_a_ = additive genetic variance; σ^2^_e_ = residual (environmental) variance, and σ^2^_p_ = phenotypic variance.

Using the BLUP model, a database and a pedigree matrix were created. Compared to the GLM method, the BLUP model is more complex, taking into account the individual animal’s genetic effects (including both direct and maternal effects) as well as maternal permanent environmental influence. This result is more accurate in estimating genetic values, especially for traits with high genetic heritability. It contains the same fixed effects as the GLM method (herd, as well as the cow’s color variant, birth year, and birth season). The BLUP model contains the pedigree matrix of animals (including the sire, mother, full and half-siblings, and grandparents), allowing for an estimate based on a whole family tree. The animal model used was as follows [29]:(3)$$y=X_{b}+Z_{u}+e$$
where y = vector of observation; b = vector of fixed effects; u = vector of random effects; e = error vector; X = incidence matrix relating observations to fixed effects, and Z = incidence matrix relating observations to random effects.

Population genetic parameters were estimated using the MTDFREML [30] program and based on the guidelines of Szőke and Komlósi [31] and Szűcs et al. [29].

### 2.4. Estimation of Genetic Values

We also estimated the BV of the sires with regard to the AFC trait, for which we used both the GLM and BLUP animal models. GLM estimates BVs based on the genetic differences between sires and does not take into account the animals’ own genetic or maternal genetic effects. The BLUP model estimates BV in more detail, taking into account direct genetic and maternal genetic effects.

The BVs were determined for all 200 sires included in this study. The BV calculation using the GLM method was performed in two steps. The calculation of the estimated progeny difference (EPD) shows how much the progeny of a given bull differs (regarding AFC) from the performance of other progeny in the population. The BV is twice the EPD. The BLUP animal model directly estimates BV.
(4)$$EPD=(x_{pg}-X_{all})$$
where x_pg_ = the mean value of the progeny group of the sire, and X_all_ = the mean value of the contemporary offspring population.

### 2.5. Phenotypic and Genetic Trends

When calculating the phenotypic trend, the AFC per year was averaged; the mean values were plotted as a function of the year of birth, and the direction and extent of the phenotypic trend were determined by linear regression analysis. The dependent variable (Y) is the assessed trait (average AFC), and the independent variable (X) is the year of birth of the cow. We determined the slope (b), which indicates the magnitude of the change and direction of the property, the constant (a), and the fit (*R*^2^).

The genetic trend of AFC was determined similarly to a previous study by Bene et al. [16] from the average BV of tested animals in two ways, using GLM and BLUP models. The genetic trend of AFC was investigated using a linear regression method and three different sources: GLM-based BV of sires; BLUP-based BV of sires; and BLUP-based BV of the entire population born in the same year. In analyzing the trend, we averaged the BV of sires and population for each year. These annual averages formed the dependent variable in the regression analysis, while the independent variable was the year of birth of the cow. Similarly to the calculation of phenotypic trends, the constant of the regression equation (a), slope (b), and degree of fit (*R*^2^) were calculated, and their statistical reliability was evaluated.

## 3. Results and Discussion

### 3.1. The Impact of Environmental Factors

The descriptive statistics for AFC are shown in Table 4. The overall mean AFC of Angus cows in Hungary was 28.1 ± 0.1 months (SD = 5.3 months, CV = 18.9%). This average is 5.02 months higher than the 33.12 (CV = 23.91%) months found in the Angus breed study by Dákay et al. [11] but lags behind the 22–24 months recommended in the literature by Day and Nogueria [6], López-Paredes et al. [5], Byrne et al. [7], Hickson et al. [4], Giess et al. [12], Bormann and Wilson [20], and Brzáková et al. [13]. In Ireland, the average AFC in heifers is 32 months, with only 23% of the population calving for the first time at the age of 24 months [7].

The average breeding age of Angus heifers in this study was 18.6 months, calculated by subtracting the mean gestation length in cattle (9.5 months) from the average AFC of 28.1 months. This value was 5–7 months lower than the optimal breeding age of 12–15 months [6].

Among the factors we examined, those of the sire (*p* < 0.01), herd (*p* < 0.05), year of birth of the cow (*p* < 0.01), and cow’s calving cycle (*p* < 0.01) had significant effects on AFC, but the color variant of the cow showed no significance (Table 5).

The Pearson correlation analysis showed a weak but significant correlation of AFC with the herd (r = 0.104, *p* < 0.01), a negative correlation with color variant (r = −0.108, *p* < 0.01), and a weak positive correlation with birth season (r = 0.060, *p* < 0.01), while the correlation with birth year was weakly negative (r = −0.063, *p* < 0.01).

The proportions of the variance in determining AFC for the factors were as follows: birth season of the cow (28.99%); birth year of the cow (28.7%); sire of the cow (18.32%); herd (11.77%); and residual (4.12%). The birth season and the birth year of the cow were the most significant factors, together explaining nearly 60% of the difference in AFC. It could be due to significant changes in environmental factors or breeding and feeding practices, weather, and grazing. The influence of the sire on the phenotypic variance in their offspring’s AFC was 18.32%, which includes both genetic and environmental effects, as confirmed by the heritability values. When examining Hungarian Limousin heifers, Bene et al. [16] found that the herd was the most influential environmental factor, accounting for 73.51% of the variance, followed by the cow’s year of birth (13.02%), the sire (6.74%), and the calving season (1.62%).

The effects of the environmental factors on AFC are summarized in Table 6. The estimated adjusted mean AFC using the GLM method was 28.3 ± 0.4 months. However, the average AFCs of the studied herds were different. In herd number 2, AFC (30.0 ± 0.7) deviated from the average by +1.7 months, while that of heifers in herd 3 (26.8 ± 0.7) deviated from the average by −1.5 months (i.e., there was a difference of 3.2 months in the AFCs of the two herds). Herd 2 is a red type of British Angus, while herd 3 is a black Angus population of the traditional British type.

The AFC was 24.9 months in 1999 but 31.6 months in 2012, with a maximum between-year difference of 6.7 months. Such differences highlight that the environmental or management conditions experienced in different years significantly influence the reproductive capacity of the cows. For example, a comparison of weather data from the Hungarian Central Statistical Office [32] for 1999 and 2012, the years producing the lowest and highest mean AFCs, showed that the average mean temperature was 10.6 and 13.0 °C, respectively. Similarly, total precipitation also differed (804 and 384 mm), as did the number of rainy days (145 and 98), the number of frosty days (103 and 63), and the number of days affected by a heat wave (1 and 33).

The AFC of spring-born individuals (27.7 ± 0.3 months) was 1.3 months earlier than those born in summer and 0.8 months later than those born in autumn. According to Bene et al. [16], one possible cause of these seasonal differences is management and husbandry, which can be related to the herds being driven to pasture in May and turned in at the end of autumn, as well as the summer drought periods. The significant effect of the calving/birth period on some reproductive traits can be attributed to seasonal changes in climatic conditions and feeding methods [33]. The age of the first calving is also influenced by the calving period of the females [13].

### 3.2. Population Genetic Parameters

The h^2^ values (Table 7) of the AFC trait (GLM 0.51 ± 0.06 and BLUP 0.38 ± 0.05) were found to be higher compared to the literature sources. This discrepancy may be attributed to differences in population structure, selection intensity, or environmental conditions. For instance, the intensive genetic selection within the Angus population studied may have amplified additive genetic variance. Additionally, the inclusion of advanced statistical models like BLUP may provide more accurate partitioning of variance components. The difference in heritability estimates between GLM (0.51) and BLUP (0.38) reflects the methodological differences in variance component estimation. As seen in Table 7, the GLM model for BLUP brings overestimated variance components and heritability values. The GLM provides a general overview; the BLUP incorporates pedigree data and corrects for relatedness, potentially yielding more accurate but lower estimates when non-additive or environmental effects are accounted for. These findings emphasize the need for a holistic approach that integrates genetic selection with optimized herd management practices to achieve long-term progress in reducing AFC. Based on data from the most literature references, the heritability of AFC trait is low (0.08 Bene et al. [16]; 0.08 Pardo et al. [34]; 0.10 Boligon and Albuquerque [18]; 0.14 Koots et al. [35]; 0.14 Giess et al. [12]; 0.17 Brzáková et al. [13]; 0.18 Zsuppán et al. [15] or moderate 0.24 Gutiérrez et al. [36]; 0.28 Bormann and Wilson [20]; 0.31 Berry and Evans [14]; 0.37 González-Murray et al. [19]; 0.46 Magaña and Segura [17]).

The correlation between direct and maternal genetic effects was strong and negative (r_dm_ = −0.97 ± 1.00). However, the standard error (SE) of 1.0 indicates that this correlation is not statistically different from zero. The majority of the variance was influenced by genetic and other non-permanent environmental factors in the Angus populations we studied. The maternal genetic influence did not have a strong influence on AFC.

### 3.3. The Influence of the Sire on AFC

The breeding values (BVs) were determined for all 200 sires included in this study, although only the data of the 15 sires with the most offspring are presented in Table 8.

Using the GLM method, we found significant differences between the mean AFC values of the juvenile groups of sires. The AFC of the sire progeny groups differs significantly. The progeny of sire registration number 20,716 calved at an average age of 31.7 ± 1.1 months (BV_GLM_ = +6.8 months), while those of sire number 20,495 calved at 22.4 ± 1.0 months (BV_GLM_ = −11.7 months), a difference of 9.3 months. We also found a large difference between the progeny groups and the BV of the sire’s AFC.

Using the BLUP animal model, the two extremes of BV (27,934: +6.0 months and 20,495: −6.2 months) produced a significant difference of 12.2 months. Bene et al. [16] found a similar difference of 15.6 months in Limousin cattle. In the BLUP model, the difference between the extreme values (the BV value) was smaller than that produced by the GLM method, but this difference was not statistically significant.

The Spearman’s rank correlation coefficient (r_s_ = 0.86; *p* < 0.01) indicates a relatively strong general agreement between the two methods. However, this correlation might not fully capture differences in rankings for sires with fewer offspring or those with extreme values. The result highlights the potential benefit of incorporating more detailed genetic information—such as individual-level data and maternal contributions—to refine the model and improve the accuracy of BV estimations.

### 3.4. Phenotypic and Genetic Trends in AFC

The calculated phenotypic and genetic trends are presented in Table 9 and Figure 1. As the table shows, only the BV of sires using the GLM method resulted in a significant change. The other effects on the trend were not statistically significant.

Based on these data, the phenotypic change was minimal (b = +0.03 ± 0.05) and non-significant (fit (*R*^2^ = 0.02; *p* > 0.05)), indicating that environmental and genetic influences and changes over time did not significantly affect the AFC.

Using GLM, the genetic trend calculated on the basis of sires decreased by −0.2 months per year (b = −0.2 ± 0.08 months; *R*^2^ = 0.18, *p* < 0.05); however, it was not significant for the BLUP animal model (b = 0.01 ± 0.03 months, *R*^2^ = 0.0, *p* > 0.05), indicating no significant change. Based on estimated direct and maternal BV in the overall population, the AFC showed no change in genetic trend (GAA_d_; b = +0.00 ± 0.01 months, *R*^2^ = 0.00, *p* > 0.05; GAA_m_; b = −0.00 ± 0.00 months, *R*^2^ = 0.00, *p* > 0.05). These results indicate that there were no significant phenotypic or genetic changes in AFC over the period we analyzed. The strong negative correlation between direct and maternal genetic effects (r_dm_ = −0.97) suggests that improving direct traits may negatively affect maternal traits, such as calving ease and progeny-rearing ability. This could create challenges in aligning breeding objectives. To develop sustainable breeding strategies, it is crucial to apply selection methods that account for both genetic effects while maintaining genetic diversity. Further research is needed to better understand this relationship and optimize breeding practices.

Although domestic research is not available on the cost–benefit analysis of AFC, our results highlight its potential economic impacts. Reducing AFC can reduce rearing costs, increase the number of calves per lifetime, and improve overall profitability. These results are consistent with international studies that have shown that earlier AFC reduces production costs and the ecological footprint.

To achieve optimal AFC, we recommend targeted genetic selection and improved herd management practices adapted to Hungarian production systems. Breeders should consider using selection indices that include AFC to improve economic performance and sustainability.

This study can provide a basis for further investigation of the economic role of AFC in Hungarian herds. Future research should focus on detailed cost–benefit analyses, the environmental impacts of AFC reduction, and the integration of AFC-related traits into breeding programs to support long-term sustainability.

## 4. Conclusions

Based on our study, the main influence on the AFC of Angus cows was the cows’ calving cycles. This result calls attention to the importance of choosing the appropriate season for calving. The stagnant phenotypic and genetic trends of AFC may also be partly explained by its potentially low genetic correlations with traits currently under direct genetic improvement, limiting the extent of indirect responses to selection. Breeders should pay more attention to this trait in the future, considering its economic relevance and potential impact on reproductive efficiency.

The differences between herds and the relatively high heritability of AFC provide an opportunity to improve this trait on a genetic basis through the development of targeted breeding programs. To achieve this, AFC could be treated as a specific selection criterion in breeding programs, with careful consideration of its inclusion alongside other traits. For instance, traits with higher genetic correlations to AFC, such as reproductive efficiency or longevity, could be prioritized in multi-trait selection indices. Conversely, traits with minimal or negative genetic correlations to AFC may require separate handling to avoid compromising progress in other key areas.

The results for the Hungarian population are determined by environmental conditions (climate, husbandry, feeding technology, genetics). The obtained results may contribute to the improvement of the AFC trait through genetic selection and management. With international cooperation, the obtained results could be comparable and extended for cross-border breeding programs.

## Figures and Tables

**Figure 1 animals-14-03715-f001:**
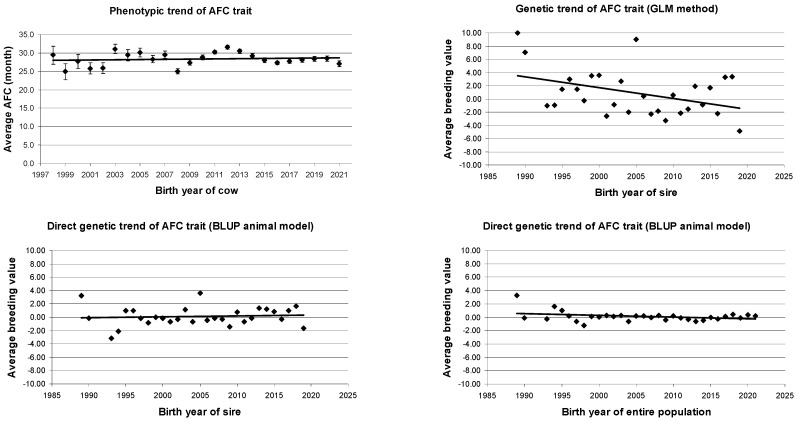
Phenotypic and genetic trends of AFC trait of Angus cows.

**Table 1 animals-14-03715-t001:** Typical mean and heritability estimates of AFC for beef cows in the literature.

Breed	AFC (Month)	h^2^	Source
ABA, HER	33.12–24.96	–	Dákay et al. [11]
RED, SIM	23.79–23.81	0.14–0.19	Giess et al. [12]
ABA, CHA	24.85–35.41	0.17–0.23	Brzáková et al. [13]
CRO	30.75	0.31	Berry and Evans [14]
LIM	33.60	0.18	Zsuppán et al. [15]
LIM	33.90	0.08	Bene et al. [16]
ZEB	34.83	0.46	Magaña and Segura [17]
NEL	34.71	0.10	Boligon and Albuquerque [18]
CRO ZEB	44.13	0.37	González-Murray et al. [19]
ABA	24.28 (AFC30)	0.28	Bormann and Wilson [20]

ABA = Aberdeen Angus; SIM = Simmental; CHA = Charolais; CRO = crossed; HER = Hereford; LIM = Limousin; RED = Red Angus; NEL = Nellore; ZEB = Zebu; h^2^ = heritability.

**Table 2 animals-14-03715-t002:** The structure of the initial database.

Starting Parameters	Database Used
Period examined, based on the cows’ birth date	1998–2021
Number of herds	5
Number of cows	2955
Number of the examined sires (sire of cow)	200
Minimum number of female progeny per sire	5
The average number of female progeny (cow) per sire	15.0
Number of the examined dams (dam of cow)	2106

**Table 3 animals-14-03715-t003:** The applied models for the estimations.

Type of Model	GLM Method	BLUP Animal Model
Random effects:		
– sire (sire of the cow);	+	−
– cow (animal);	−	+
– maternal genetic effect.	−	+
Fixed effects:		
– herd;	+	+
– color variant of the cow;	+	+
– birth year of the cow;	+	+
– birth season of the cow.	+	+
Pedigree matrix:		
– animal (cow);	−	+
– sire;	−	+
– dam;	−	+
– full sibs, half sibs;	−	+
– grandparents.	−	+
Examined trait:		
– age at first calving.	+	+

+, the model included this effect; −, the model did not include this effect.

**Table 4 animals-14-03715-t004:** Descriptive statistics of the age at first calving in Angus cows.

Parameters	Age at First Calving
*n*	2955
Mean (months)	28.1
Standard error (S.E.) (months)	0.1
Standard deviation (SD) (months)	5.3
Coefficient of variation (CV, %)	18.9
Median (months)	25.9
Minimum (months)	19
Maximum (months)	46
Kolgomorov–Smirnov test (*p*)	0.00

**Table 5 animals-14-03715-t005:** The effect of different factors on the age at first calving.

Trait	Classes	Age at First Calving	
Factor		Effect (*p*)	Rate in Phenotype (%)
Sire of cow	200	<0.01	18.32
Herd	5	<0.05	11.77
Color variant of the cow	2	*>0.05*	8.10
Birth year of the cow	24	<0.01	28.70
Birth season of the cow	4	<0.01	28.99
Residual	–	–	4.12
Total	–	–	100.00

**Table 6 animals-14-03715-t006:** The effect of environmental factors on the age at first calving.

Trait	*n*	Age at First Calving (Months)	
Adjusted Overall Mean (±SE)	2955	28.3 ± 0.3	
Environmental Factors		Mean ± SE	Deviation from the Overall Mean
Herd (code)			
1	243	29.0 ± 0.7	+0.7
2	710	30.0 ± 0.7	+1.7
3	93	26.8 ± 0.7	−1.5
4	1266	27.6 ± 0.4	−0.7
5	643	28.2 ± 0.5	−0.1
Color variant of the cow			
Black	1445	28.5 ± 0.4	+0.2
Red	1510	28.0 ± 0.4	−0.3
Birth year of the cow			
1998	25	29.4 ± 2.4	+1.1
1999	18	24.9 ± 2.2	−3.4
2000	15	27.7 ± 1.9	−0.6
2001	36	25.8 ± 1.5	−2.5
2002	80	25.9 ± 1.5	−2.4
2003	37	31.1 ± 1.3	+2.8
2004	42	29.4 ± 1.5	+1.1
2005	37	30.1 ± 1.2	+1.8
2006	102	28.3 ± 1.0	+0.0
2007	60	29.5 ± 1.0	+1.2
2008	94	25.0 ± 0.7	−3.3
2009	113	27.4 ± 0.7	−0.9
2010	109	28.8 ± 0.7	+0.5
2011	197	30.2 ± 0.5	+1.9
2012	215	31.6 ± 0.5	+3.3
2013	187	30.5 ± 0.6	+2.2
2014	169	29.2 ± 0.6	+0.9
2015	201	28.0 ± 0.6	−0.3
2016	315	27.3 ± 0.5	−1.0
2017	252	27.8 ± 0.6	−0.5
2018	196	28.1 ± 0.6	−0.2
2019	160	28.4 ± 0.7	+0.1
2020	178	28.5 ± 0.7	+0.2
2021	117	27.1 ± 0.8	−1.2
Birth season of the cow			
Winter	464	28.1 ± 0.4	−0.2
Spring	1759	27.7 ± 0.3	−0.6
Summer	550	29.0 ± 0.4	+0.7
Autumn	182	28.5 ± 0.5	+0.2

Herd code: 1 = Canadian, red; 2 = British, red; 3 = British, black; 4 = British, red; 5 = other.

**Table 7 animals-14-03715-t007:** Population genetic parameters of the age at first calving trait.

Parameters	Age at First Calving	
	GLM Method	BLUP Animal Model
σ^2^_s_; σ^2^_a_	21.49	9.77
σ^2^_m_	–	0.00
σ_dm_	–	−0.01
σ^2^_e_	20.94	15.98
σ^2^_p_	42.43	25.74
h^2^_d_	0.51 ± 0.06	0.38 ± 0.05
h^2^_m_	–	0.00 ± 0.03
r_dm_	–	−0.97 ± 1.00
e^2^	–	0.62 ± 0.05

σ^2^_s_ = sire variance in GLM; σ^2^_m_ = maternal genetic variance; σ_dm_ = direct maternal genetic covariance; σ^2^_e_ = residual variance; σ^2^_p_ = phenotypic variance; h^2^_d_ = direct heritability; h^2^_m_ = maternal heritability; r_dm_ = direct–maternal genetic correlation; e^2^ = the ratio of the residual variance to the phenotypic variance.

**Table 8 animals-14-03715-t008:** The effect of sire on the age at first calving trait of Angus cows.

Trait	*n*	Age at First Calving (Month)			
Sire of Cow (Registration Number) _#_		GLM Method		BLUP Animal Model	
		Mean of Progeny (±SE)	BV_GLM_	BV_AMD_	BV_AMM_
Corrected mean (±SE)	2955	28.3 ± 0.3			
20,495	84	22.4 ± 1.0	−11.7	−6.2	+0.0
20,501	40	29.2 ± 1.0	+1.8	−0.9	+0.0
20,716	41	31.7 ± 1.0	+6.8	+4.8	−0.0
22,666	50	24.1 ± 1.0	−8.5	−2.7	+0.0
23,155	40	27.1 ± 1.0	−2.3	−2.2	+0.0
24,100	39	28.9 ± 1.0	+1.3	−0.8	+0.0
24,608	68	29.5 ± 1.0	+2.4	+0.8	−0.0
24,626	44	30.6 ± 1.0	+4.6	+3.6	−0.0
24,635	42	22.4 ± 1.1	−11.7	−5.9	+0.0
27,934	35	29.4 ± 1.2	+2.2	+6.0	−0.0
27,940	56	28.0 ± 1.1	−0.7	+3.2	−0.0
27,946	56	28.6 ± 1.1	+0.7	+3.6	−0.0
30,947	61	27.2 ± 1.0	−2.3	−1.6	+0.0
31,117	68	28.0 ± 1.2	−0.6	+1.5	−0.0
34,296	35	28.7 ± 1.2	+0.9	+0.7	−0.0
r_s_	BV_GLM_ BV_AMD_			+0.86 *	−0.86 * −0.99 *

^#^ breeding values are presented only for the 15 sires with the most offspring; * *p* < 0.01; *n* = number of progeny; BV_GLM_ = breeding value estimated with GLM method; BV_AMD_ = direct breeding value estimated with BLUP animal model; BV_AMM_ = maternal breeding value estimated with BLUP animal model; r_s_ = Spearman’s rank correlation coefficient.

**Table 9 animals-14-03715-t009:** Phenotypic and genetic trends in age at first calving.

Trend	Y	b	SE	*p*	a	SE	*p*	*R* ^2^	*p*
P	AFC	+0.03	0.05	*>0.05*	−37.11	107.84	*>0.05*	0.02	*>0.05*
GGS	AFC^BV^	−0.20	0.08	<0.05	403.27	163.47	<0.05	0.18	<0.05
GAS_d_	AFC^BV^	+0.01	0.03	*>0.05*	−11.40	64.42	*>0.05*	0.00	*>0.05*
GAS_m_	AFC^BV^	−0.00	0.00	*>0.05*	0.02	0.08	*>0.05*	0.00	*>0.05*
GAA_d_	AFC^BV^	+0.00	0.01	*>0.05*	−0.63	21.89	*>0.05*	0.00	*>0.05*
GAA_m_	AFC^BV^	−0.00	0.00	*>0.05*	0.00	0.03	*>0.05*	0.00	*>0.05*

P = phenotypic trend; GGS = genetic trend in BV of sires using the GLM method; GAS_d_ = genetic trend in direct BV of sires using the BLUP model; GAS_m_ = genetic trend in maternal BV of sires using the BLUP model; GAA_d_ = genetic trend in direct BV of all animals using the BLUP model; GAA_m_ = genetic trend in maternal BV of all animals using the BLUP model; X = birth year; AFC = average age at first calving (months); AFC^BV^ = average breeding value in age at first calving trait (months); BV = breeding value.

## Data Availability

The data presented in this study are available on request from the Hungarian Hereford, Angus, and Galloway Breeders’ Association.

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
