# Peer review of "Heritability Estimates of Age at First Calving and Correlation Analysis in Angus Cows Bred in Hungary"

_animals, 2024, doi:10.3390/ani14243715_

Round 1
Reviewer 1 Report
Comments and Suggestions for Authors
Dear Authors, I have deeply revised your paper and I found the topic really interesting and important for the breeding management of beef cattle. The methodology is also appropriate, apart for some lacks in the analysis and also in the explanation and interpretation of the results. Further details are reported in my detailed comments. I strongly suggest to fix all the issues I raised, otherwise I think the paper should have too serious flaws to be considered suitable for publication.
Detailed comments:
L15: previously reported in literature or in other studies or technical reports on the same breeds?
L21: It Is not clear if you are using the genetic parameters as phenotypes - and this should be wrong
L28: the cow’s colour variant (8.10%, NS). This Is a non significante effect in the model
L32: You should not have the Pe in the model, since you have single records per animal. Please, remove this effect from the model and run again the analysis
L53: could you ad a reference?
L109: “... in five different genetic groups”: Table 2 is not reporting the genetic groups
L112: (miscarriages and stillborn calves were not considered): why they are not considered? you could introduce a bias in such a way. What Is the incidence of miscarriages and stillborn animals?
L115: What kind of color is this one? Blue, yellow,red and green: Is there a specific meaning for these colors?
L126-128: you should specify at the end of the sentence “before running the genetic analysis using the BLUP animal model”. Moreover: why did you include the sire instead of the cow?
L140-141: I suggest to delete “entity” for a matter of simplicity
L143: Looking at Table 3 I can see that the GLM includes the sire variance, not the additive genetic variance, that Is used in the BLUP. Is It correct? Could you correct the sentence?
L151: Again, reading Table 3 the sire Is not included in BLUP
L159: I suggest to specify that X and Z are incidence matrices
L165: GLM estimates the BV
L165: phenotypic, not genetic differences among sires
L169-170: I suggest to move the sentence in brackets within the Results
L198-199: Ref 9: Could you specify in which breed?
L217: this Is not a genetic influence of the sire but a general influence including both genetic and environmental effects
L225: Some numbers differ between the text and Table 6 (the mean of herds number 1 and 4)
L230 Table 6: I suggest to report the Birth year as a graph, the trend should be clearer. Again, while reporting the results, take into account that n is low in the oldest birth years, and therefore the SE is higher
L259: Table 7: The one reported via GLM is not direct heritability, it is just the sire effect, that embraces additive genetic and permanent environemntal components, as I reported above. In some studies it is called the repeatability of a trait, that is the upper limit of the heritability
L265: a correlation with a SE of 1 is not significantly different from zero. Please, report this information
L266-267: It is quite obvious, since you do not have repeated information per individual. I suggest to rerun the analysis by deleting this effect, and to adjust all the results accordingly. I also think that deleting this effect also the estimation of the other genetic effects may improve.
L284: A spearman rank coefficient of 0.86 is not so strong when you look at the top individuals of a population, like the top sires, therefore it is not really true that the two methods reveal a similar genetic pattern. However, the information that you could obtain from this result is that the inclusion of more detailed genetic information, coming from the individual (not sire) and from the mother, can add relevant information to the model and therefore leading to a partial reranking of the BVs.
L313 and 314: the trend should be stagnant also due to possible low genetic correlations with traits under direct genetic improvement (no indirect response to selection). I suggest to add a consideration on this point. Moreover, you should add some more consideration about how to treat the trait from a genetic improvement point of view, therefore by mentioning what traits you suggest to include or not
Author Response
Cover letter
Manuscript ID: animals-3318077
Type of manuscript: Article
Title: Population genetic characteristics of the age at first calving of Angus cows bred in Hungary
Authors: Judit Márton, Szabolcs Albin Bene and Ferenc Szabó
Response to the Reviewer 1
Dear Reviewer, Thank you for your careful review of our work and your comments. We have corrected the issues raised, which we hope will contribute to making our paper suitable for publication.
Detailed comments:
Comment: L15: previously reported in literature or in other studies or technical reports on the same breeds?
Response: Yes, in terms of literature references on the Aberdeen Angus breed in terms of AFC, for example: (the missing Giess et al. (2022) literature has been inserted into the text)
- h2 = 0.175 (Brzáková, M. - ÄŒítek, J. - Svitáková, A. - Veselá, Z. - Vostrý, L.: Genetic parameters for age at first calving and first calving interval of beef cattle. Animals, 2020, 10, 2122. https://org/10.3390/ani10112122)
- h2 = 28 and AFC = 24.28 month (Bormann J.M. - Wilson D.E.: Calving day and age at first calving in Angus heifers. J. Anim. Sci., 2010, 88, 1947–1956. https://doi.org/10.2527/jas.2009-2249)
- h2 = 0.14 and AFC = 23.79 month (Giess, L.K. - Boldt, R.J. - Culbertson, M. - Thomas, M.G. - Speidel, S.E. - Enns, R.M.: 16 genetic parameter estimates for age at first calving in Simmental and Red Angus heifers. J. Anim. Sci., 2022, 100, S4, 9–10, https://doi.org/10.1093/jas/skac313.012)
Comment: L21: It is not clear if you are using the genetic parameters as phenotypes - and this should be wrong
Response: Thank you for raising this important point. In our study, genetic parameters such as heritability and breeding values were not treated as phenotypes. Instead, the age at first calving (AFC) was considered as a phenotypic trait, defined as the difference between the cow’s date of birth and the date of first calving. This phenotypic data was used as the basis for estimating genetic parameters through GLM and BLUP models.
The genetic parameters were derived from the observed phenotypic variance components of AFC, including additive genetic variance, environmental variance, and total phenotypic variance. These derived parameters were modelled following established methodologies and were not analysed as direct phenotypic traits themselves.
This approach aligns with standard practices in quantitative genetic analysis, as outlined in SzÅ‘ke and Komlósi [32] and Szűcs et al. [30], ensuring that the genetic parameters were statistically derived rather than treated as observable traits.
Comment: L28: the cow’s colour variant (8.10%, NS). This is a non-significant effect in the model
Response: After the sentence in the abstract, we made the following addition: The cow’s colour variant was not significant and did not influence the AFC in this study. (L28-29)
Comment: L32: You should not have the Pe in the model, since you have single records per animal. Please, remove this effect from the model and run again the analysis
Response: Thank you for your comment, we agree with it. We have removed “Pe” from the study.
Comment: L53: could you ad a reference?
Response: I apologize for the inattention. I have added the references to the literature at the end of the sentence and also included them in the References section. (L52-53; L409-414)
The references are:
- ShortE. - Staigmiller R.B. - Bellows R.A. - Greer R.C.: Breeding heifers at one year of age: Biological and economic considerations. In: Fields M.J. - Sands R.S., editors, Factors affecting calf crop. CRC Press, Boca Raton, 1994. 55–68. https://doi.org/10.1201/9781003069119-4
- Hickson, R.E. - Lopez-Villalobos, N. - Kenyon, P.R. - Ridler, B.J. - Morris, S.T. Profitability of calving heifers at 2 compared with 3 years of age and the effect of incidence of assistance at parturition on profitability. Anim. Prod. Sci., 2010, 50, 354. https://doi.org/10.1071/an09180
Comment: L109: “... in five different genetic groups”: Table 2 is not reporting the genetic groups
Response: Thanks for the comment, I have corrected the sentence as follows: The dataset included 2,955 pedigree Angus and high (>75%) blood Angus cows (Table 2) in five different genetic groups [26]. (L116-118)
Comment: L112: (miscarriages and stillborn calves were not considered): why they are not considered? you could introduce a bias in such a way. What Is the incidence of miscarriages and stillborn animals?
Response: In a significant number of farms, miscarriages and stillbirths are not reliably documented. Therefore, these data were not considered, even for the farms that do track them. According to estimates from the Hungarian Hereford, Angus, and Galloway Breeders’ Association, the abortion rate is 2%, while the occurrence of stillbirths remains under 5%.
Comment: L115: What kind of color is this one? Blue, yellow, red and green: Is there a specific meaning for these colors?
Response: The colours are the designations used for genetic groups in the previous study cited [26]. In the figures, colours classified into different genetic groups are displayed with these colors. For clarity, we have removed the colours from the study and used the numbers to denote the groups. (L126-130)
Comment: L126-128: you should specify at the end of the sentence “before running the genetic analysis using the BLUP animal model”. Moreover: why did you include the sire instead of the cow?
Response: Adding clarification to the GLM description: We appreciate your suggestion and agree that the sentence should be clarified. The revised text will state: The effects of fixed and random factors influencing the trait AFC were evaluated using a general linear model (GLM) within ANOVA Type III univariate analysis of variance (Table 3), before running the genetic analysis using the BLUP animal model. (L139-151)
This ensures the methodological sequence is clear.
Rationale for including sire instead of cow in the GLM model: The decision to include the sire as a random effect in the GLM model rather than the cow is due to the limitations of this method. GLM does not account for the full pedigree structure or the genetic relationships between individuals, as it lacks an animal model framework. Including the sire allows us to capture the variance attributable to paternal genetic effects, which provides an initial understanding of genetic influences. The BLUP animal model, applied subsequently, accounts for the genetic contributions of both sire and dam along with the maternal environmental effects, offering a more precise estimation.
Comment: L140-141: I suggest to delete “entity” for a matter of simplicity
Response: The sentence has been corrected accordingly: The available database allowed for both a simpler sire model and a more complex animal model. Population genetic parameters were determined using the GLM [17] and BLUP (Best Linear Unbiased Prediction) [28] animal models. (L153-155)
Comment: L143: Looking at Table 3 I can see that the GLM includes the sire variance, not the additive genetic variance, that Is used in the BLUP. Is It correct? Could you correct the sentence?
Response: The sentence has been corrected accordingly. The population genetic parameters determined by the GLM model were: σ²s = sire variance, σ²e = residual variance, σ²p = phenotypic variance and h²s = heritability estimated based on sire variance. This differs from the BLUP model, which directly estimates additive genetic variance. (L156-159)
Comment: L151: Again, reading Table 3 the sire Is not included in BLUP
Response: The error in the text has been corrected: Using the BLUP model, a database and a pedigree matrix were created. Compared to the GLM method, the BLUP model is more complex, taking into account the individual animal's genetic effects (including both direct and maternal effects) as well as maternal permanent environmental influence. (L164-167)
Comment: L159: I suggest to specify that X and Z are incidence matrices
Response: Thanks for the clarification. The study has been corrected accordingly. X = incidence matrix relating observations to fixed effects, and Z = incidence matrix relating observations to random effects. (L173-175)
Comment: L165: GLM estimates the BV; L165: phenotypic, not genetic differences among sires
Response: We clarified in the text that the estimation of BV (breeding value) based on GLM is based on phenotypic differences as follows: We also estimated the BV of the sires with regard to the AFC trait using both the GLM and BLUP animal models. GLM estimates BVs based on phenotypic differences among sires and does not take into account the animals’ own genetic or maternal genetic effects. In contrast, the BLUP model estimates BV in more detail, considering both direct genetic and maternal genetic effects. (L179-183)
Comment: L169-170: I suggest to move the sentence in brackets within the Results
Response: The note regarding the table has been moved to the sentence referring to the results section as suggested as follows: The BVs were determined for all 200 sires included in the study. The BV calculation using the GLM method was performed in two steps. (Details regarding the 15 sires with the most offspring are provided in Table 8, as part of the Results section.) (L184-186)
Comment: L198-199: Ref 9: Could you specify in which breed?
Response: This data from the study by Dákay et al. refers to the Angus breed. In the study, we have supplemented the text with this as follows: This average is 5.02 months higher than the 33.12 months found in the Angus breed study by Dákay et al. [12]. (L213-215)
Comment: L217: this Is not a genetic influence of the sire but a general influence including both genetic and environmental effects
Response: Thank you, we have corrected the text accordingly as follows: The influence of the sire on the phenotypic variance in their offspring’s age at first calving (AFC) was 18.32%, which includes both genetic and environmental effects, as confirmed by the heritability values. (L238-240)
Comment: L225: Some numbers differ between the text and Table 6 (the mean of herds number 1 and 4)
Response: Of course we have corrected it, thank you for your comment: However, the average AFCs of the studied herds were different. In herd number 2, AFC (30.0±0.7) deviated from the average by +1.7 months, while that of heifers in herd 3 (26.8±0.7) deviated from the average by –1.5 months (i.e. there was a difference of 3.2 months in the AFCs of the two herds). Herd 2 is a red type of British Angus, while herd 3 is black Angus population of the traditional British type. (L245-250)
Comment: L230 Table 6: I suggest to report the Birth year as a graph, the trend should be clearer. Again, while reporting the results, take into account that n is low in the oldest birth years, and therefore the SE is higher
Response: Thank you for the suggestion, the figure (Figure 1) has been included in the study. (L353)
Figure 1: Phenotypic and genetic trends of AFC trait of Angus cows
Comment: L259: Table 7: The one reported via GLM is not direct heritability, it is just the sire effect, that embraces additive genetic and permanent environmental components, as I reported above. In some studies it is called the repeatability of a trait, that is the upper limit of the heritability
Response: We have added the following addition to the table: σ²s = sire variance in GLM. (L290-294)
Comment: L265: a correlation with a SE of 1 is not significantly different from zero. Please, report this information
Response: We have added the following addition after the sentence: However, the standard error (SE) of 1.0 indicates that this correlation is not statistically different from zero. (L296-297)
Comment: L266-267: It is quite obvious, since you do not have repeated information per individual. I suggest to rerun the analysis by deleting this effect, and to adjust all the results accordingly. I also think that deleting this effect also the estimation of the other genetic effects may improve.
Response: Thank you for your comment, we agree with it. We have removed “Pe” from the study.
Comment: L284: A spearman rank coefficient of 0.86 is not so strong when you look at the top individuals of a population, like the top sires, therefore it is not really true that the two methods reveal a similar genetic pattern. However, the information that you could obtain from this result is that the inclusion of more detailed genetic information, coming from the individual (not sire) and from the mother, can add relevant information to the model and therefore leading to a partial reranking of the BVs.
Response: The text has been modified to take into account the recommendation: The breeding values (BVs) were determined for all 200 sires included in the study, although only the data of the 15 sires with the most offspring are presented in Table 8. (L301-302)
The Spearman’s rank correlation coefficient (rs = 0.86; p<0.01) indicates a relatively strong general agreement between the two methods. However, this correlation might not fully capture differences in rankings for sires with fewer offspring or those with extreme values. The result highlights the potential benefit of incorporating more detailed genetic information - such as individual-level data and maternal contributions - to refine the model and improve the accuracy of BV estimations. (L314-319)
Comment: L313 and 314: the trend should be stagnant also due to possible low genetic correlations with traits under direct genetic improvement (no indirect response to selection). I suggest to add a consideration on this point. Moreover, you should add some more consideration about how to treat the trait from a genetic improvement point of view, therefore by mentioning what traits you suggest to include or not
Response: Thank you for your comment. We have revised the conclusions accordingly.
Based on our study, the main influence on the AFC of Angus cows was the cows’ calving cycle. This result calls attention to the importance of choosing the appropriate season for calving. The stagnant phenotypic and genetic trends of AFC may also be partly explained by its potentially low genetic correlations with traits currently under direct genetic improvement, limiting the extent of indirect responses to selection. Breeders should pay more attention to this trait in the future, considering its economic relevance and potential impact on reproductive efficiency.
The differences between herds and the relatively high heritability of AFC provide an opportunity to improve this trait on a genetic basis through the development of targeted breeding programs. To achieve this, AFC could be treated as a specific selection criterion in breeding programs, with careful consideration of its inclusion along-side other traits. For instance, traits with higher genetic correlations to AFC, such as reproductive efficiency or longevity, could be prioritized in multi-trait selection indices. Conversely, traits with minimal or negative genetic correlations to AFC may require separate handling to avoid compromising progress in other key areas.
The results for the Hungarian population are determined by environmental conditions (climate, husbandry and feeding technology, genetics). The obtained results may contribute to the improvement of the AFC trait through genetic selection and management. With international cooperation, the obtained results could be comparable and extended for cross-border breeding programs. (L368-387)

Reviewer 2 Report
Comments and Suggestions for Authors
I have carefully reviewed the manuscript "Population genetic characteristics of the age at first calving of Angus cows bred in Hungary." This study provides valuable insights into genetic and environmental factors affecting age at first calving (AFC) in Hungarian Angus cattle populations. Below are my detailed comments and suggestions:
Major revisions:
- Methods:
- Please clarify the criteria used for data filtering/quality control before analysis
- Provide more details about how the five genetic groups were determined
- Include justification for the choice of environmental factors in the models
- Specify software versions for all statistical analyses
- Results:
- Add confidence intervals for key parameters where appropriate
- Consider including a correlation matrix between major traits
- Provide more detailed results of model diagnostics/goodness of fit tests
- Include effect sizes alongside p-values for statistical comparisons
- Discussion:
- Expand discussion of why heritability estimates differ from previous studies
- Address potential limitations of using data from only Hungarian herds
- Compare findings more extensively with international Angus populations
- Discuss practical implications of the birth season effects found
- Tables and Figures:
- Consider adding a figure showing phenotypic/genetic trends over time
- Table 4 could benefit from additional descriptive statistics
- Add error bars or confidence intervals to relevant figures
- Consider combining Tables 5 and 6 for better data presentation
Minor revisions:
- Technical corrections:
- Line 25: Add standard deviation after "28.1 ± 0.1 months"
- Line 72: Define "SLA" at first use
- Lines 229-232: Clarify herd type descriptions
- Check consistency in decimal place reporting throughout
- Language and style:
- Improve clarity of sentences in lines 39-44
- Ensure consistent use of terminology (e.g., "age at first calving" vs "AFC")
- Review grammar in lines 208-210
- Consider shortening some lengthy paragraphs in the discussion
- References:
- Update references where more recent studies are available
- Ensure consistent formatting of DOIs
- Consider adding key references about Angus breeding in Central Europe
Additional suggestions:
- Consider adding:
- Brief description of Hungarian Angus breeding objectives
- Economic implications of the findings
- Recommendations for practical implementation
- Future research directions
- Methodological considerations:
- Discuss potential bias in data collection
- Address missing data handling
- Explain choice of statistical models more thoroughly
The manuscript makes a valuable contribution to understanding genetic parameters affecting AFC in Angus cattle. With the suggested revisions, it will provide even more valuable insights for both researchers and practitioners.
Questions whoch answers should be added to the mansucript:
- Were there any notable differences in management practices across herds that might have influenced results?
- How do these results compare specifically to other Central European Angus populations?
- What practical recommendations would you make to breeders based on these findings?
Author Response
Cover letter
Manuscript ID: animals-3318077
Type of manuscript: Article
Title: Population genetic characteristics of the age at first calving of
Angus cows bred in Hungary
Authors: Judit Márton, Szabolcs Albin Bene and Ferenc Szabó
Response to the Reviewer 2
Dear Reviewer, Thank you for your careful review of our work and your comments. We have corrected the issues raised, which we hope will contribute to making our paper suitable for publication.
- Major revisions
1.1. Methods
Comment: Please clarify the criteria used for data filtering/quality control before analysis
Response: The data filtering was done as follows after selection from the breeding association's database: Only individuals with known maternal and paternal lineage were included in the analysis. From the database, 200 cows were excluded as their records could have skewed the results.
For AFC, we removed two outliers under 19.0 months and 197 individuals over 46.0 months (first calving above 7 years). The latter were likely due to administrative errors related to the renumbering of Hungarian herds during 1997-1998.
Additionally, we did not include abortions or stillbirths, as not all 30 herds reliably recorded this data, which could have caused result distortions.
In accordance with the suggestion, the following addition was made to the text: Only individuals with known maternal and paternal lineage were included in the analysis. When determining the date of the first calving, only live calves were taken into account (aborted and stillborn calves were not considered) and a total of 199 outliers under 19.0 months (2) and over 46.0 months (179) were excluded. (L120-123)
Comment: Provide more details about how the five genetic groups were determined
Response: The grouping was based on a previous study, Márton et al. [26] (https://doi.org/10.5713/ab.23.0157). In this research, we analyzed the genetic parameters of 1,369 individuals from 16 Hungarian Angus nucleus herds using genotyping data from 12 microsatellite markers. The number of alleles per locus ranged between 11 and 18, with an average effective allele number Ne = 3.201, an expected heterozygosity He = 0.659 and an observed heterozygosity Ho = 0.710. Based on these findings, the 16 nucleus herds were categorized into four genetically distinct groups, differentiated by origin, colour, size, and type: Blue group (BG), Red group (RG), Green group (GG), Yellow group (YG): The colours are the designations used for genetic groups in the previous study cited. In the figures, cultures classified into different genetic groups are displayed with these colours. For clarity, we have removed the colours from the study and used the numbers to denote the groups. (L126-130)
The fifth group consists of 14 herds that lacked sufficient microsatellite data for inclusion in the original genetic analysis but met all other criteria for this study. The animals in these herds originate from the herds in the four genetic groups.
Comment: Include justification for the choice of environmental factors in the models
Response: The choice of environmental factors was based on our previous work: Bene et al. [17] (https://doi.org/10.5513/JCEA01/22.2.3161). Examined environmental factors are: herd, birth year of cow, birth season of cow.
Comment: Specify software versions for all statistical analyses
Response:
- The data was selected by the Hungarian Hereford, Angus, and Galloway Breeders’ Association’s own developed software (MHAGTE registry) - We have added it to the text. (L133-134)
- Data preparation was carried out using the programs Microsoft Excel and Word 2021.
- The evaluation and the correlation matrix were conducted using IBM SPSS Statistics for Windows, Version 27.0. (L135-137)
- The effects of fixed and random factors influencing the trait AFC were evaluated using a general linear model (GLM) within ANOVA Type III univariate analysis of variance
- Population genetic parameters were determined using the GLM and BLUP (Best Linear Unbiased Prediction) animal models
- Population genetic parameters were estimated using MTDFREML program
1.2. Results
Comment: Add confidence intervals for key parameters where appropriate
Response: In our own study the overall average AFC of Angus cows in Hungary was 28.1 ± 0.1 months (SD = 5.3 months, CV = 18.9%). The CV of the data in the Dákay et al. reference based on the data found in the literature reference is: CV = (0.66/2.76) *100 ≈ 23.91%, we have inserted this into the paper. Dákay et al. [12] (L213-215)
Confidence intervals can be expressed in several ways. Table 4 shows the mean and standard error (SE) and Table 6 shows the Mean±SE values. We consider that these express the confidence interval.
Comment: Consider including a correlation matrix between major traits
Response: Thank you for your comment. We have included the suggested information in the manuscript based on the matrix. The Pearson correlation analysis showed a weak but significant correlation of AFC with herd (r = 0.10, p<0.01), a negative correlation with colour variant (r = -0.108, p<0.01), and a weak positive correlation with birth season (r = 0.06, p<0.01), while the correlation with birth year was weakly negative (r = -0.06, p<0.01). (L228-231)
Correlations |
||||||
|
Herd |
Colour |
Birth_season |
AFC |
Birth_year |
|
Herd |
Pearson Correlation |
1 |
-,527** |
-,189** |
,104** |
,029 |
Sig. (2-tailed) |
|
,000 |
,000 |
,000 |
,112 |
|
N |
2955 |
2955 |
2955 |
2955 |
2955 |
|
Colour |
Pearson Correlation |
-,527** |
1 |
,220** |
-,108** |
,146** |
Sig. (2-tailed) |
,000 |
|
,000 |
,000 |
,000 |
|
N |
2955 |
2955 |
2955 |
2955 |
2955 |
|
Birth_season |
Pearson Correlation |
-,189** |
,220** |
1 |
,060** |
-,032 |
Sig. (2-tailed) |
,000 |
,000 |
|
,001 |
,084 |
|
N |
2955 |
2955 |
2955 |
2955 |
2955 |
|
AFC |
Pearson Correlation |
,104** |
-,108** |
,060** |
1 |
-,063** |
Sig. (2-tailed) |
,000 |
,000 |
,001 |
|
,001 |
|
N |
2955 |
2955 |
2955 |
2955 |
2955 |
|
Birth_year |
Pearson Correlation |
,029 |
,146** |
-,032 |
-,063** |
1 |
Sig. (2-tailed) |
,112 |
,000 |
,084 |
,001 |
|
|
N |
2955 |
2955 |
2955 |
2955 |
2955 |
|
**. Correlation is significant at the 0.01 level (2-tailed). |
Comment: Provide more detailed results of model diagnostics/goodness of fit tests:
Response: Thank you for your comment, we have included the requested information in the paper: As it is seen in Table 7 GLM model for BLUP brings overestimated variance components and heritability values. (L278-280)
Comment: Include effect sizes alongside p-values for statistical comparisons
Response: We have included the requested information in the paper: As the table shows, only the BV of sires using the GLM method resulted in a significant change. The other effects on the trend were not statistically significant. (L327-328)
1.3. Discussion
Comment: Expand discussion of why heritability estimates differ from previous studies
Response: Thanks for the comment. The study has been corrected accordingly: The h2 values (Table 7) of the AFC trait (GLM 0.51±0.06 and BLUP 0.38±0.05) were found to be higher compared to literature sources. This discrepancy may be attributed to differences in population structure, selection intensity, or environmental conditions. For instance, the intensive genetic selection within the Angus population studied may have amplified additive genetic variance. Additionally, the inclusion of advanced statistical models like BLUP may provide more accurate partitioning of variance components. The difference in heritability estimates between GLM (0.51) and BLUP (0.38) reflects the methodological differences in variance component estimation. As it is seen in Table 7 GLM model for BLUP brings overestimated variance components and heritability values. The GLM provides a general overview, BLUP incorporates pedigree data and corrects for relatedness, potentially yielding more accurate but lower estimates when non-additive or environmental effects are accounted for. These findings emphasize the need for a holistic approach that integrates genetic selection with optimized herd management practices to achieve long-term progress in reducing AFC. (L271-284)
Comment: Address potential limitations of using data from only Hungarian herds + Compare findings more extensively with international Angus populations + Discuss practical implications of the birth season effects found
Response: We have revised the conclusions accordingly.
Based on our study, the leading influence on the AFC of Angus cows was the cows’ calving cycle. This result calls attention to the importance of choosing the appropriate season for calving. The stagnant phenotypic and genetic trends of AFC may also be partly explained by its potentially low genetic correlations with traits currently under direct genetic improvement, limiting the extent of indirect responses to selection. Breeders should pay more attention to this trait in the future, considering its economic relevance and potential impact on reproductive efficiency.
The differences between herds and the relatively high heritability of AFC provide an opportunity to improve this trait on a genetic basis through the development of targeted breeding programs. To achieve this, AFC could be treated as a specific selection criterion in breeding programs, with careful consideration of its inclusion alongside other traits. For instance, traits with higher genetic correlations to AFC, such as reproductive efficiency or longevity, could be prioritized in multi-trait selection indices. Conversely, traits with minimal or negative genetic correlations to AFC may re-quire separate handling to avoid compromising progress in other key areas.
The results for the Hungarian population are determined by environmental conditions (climate, husbandry and feeding technology, genetics). The obtained results may contribute to the improvement of the AFC trait through genetic selection and management. With international cooperation, the obtained results could be comparable and extended for cross-border breeding programs. (L368-387)
1.4. Tables and Figures
Comment: Consider adding a figure showing phenotypic/genetic trends over time
Response: Thank you for the suggestion, the figure (Figure 1) has been included in the study. (L353)
Comment: Table 4 could benefit from additional descriptive statistics
Response: It is believed that the descriptive statistics presented in Table 4 are consistent with the objective of the study. The derived data are presented in the additional tables.
Comment: Add error bars or confidence intervals to relevant figures
Response: The requested data has been added.
Comment: Consider combining Tables 5 and 6 for better data presentation
Response: More explanation of the data in Tables 5 and 6 have been added to the manuscript.
- Minor revisions
2.1. Technical corrections
Comments: Line 25: Add standard deviation after "28.1 ± 0.1 months"
Response: The average AFC obtained was 28.1±0.1 months (SD = 5.3 months), showing a relatively large variance (CV = 18.9%). (L25-26)
Comment: Line 72: Define "SLA" at first use.
Response: The SLA was removed from the manuscript.
Comment: Lines 229-232: Clarify herd type descriptions:
Response: Of course we have corrected it, thank you for your comment. However, the average AFCs of the studied herds were different. In herd number 2, AFC (30.0±0.7) deviated from the average by +1.7 months, while that of heifers in herd 3 (26.8±0.7) deviated from the average by –1.5 months (i.e. there was a difference of 3.2 months in the AFCs of the two herds). Herd 2 is a red type of British Angus, while herd 3 is black Angus population of the traditional British type. (L245-250)
Comment: Check consistency in decimal place reporting throughout:
Response: Decimals have been checked and modified.
2.2. Language and style
Comment: Improve clarity of sentences in lines 39-44
Response: Thank you for your comment. We have revised the sentences to make them clearer and easier to understand. The revised text now reads as follows: The profitability and sustainability of beef herds are significantly influenced by the age at first calving (AFC), which has a direct impact on the reproductive and productive performance of the cows. Regarding the economic sustainability of beef cattle farming, reproductive performance is a decisive factor. According to Pulina et al. [1], the main limitations in the beef sector include unfavourable reproduction, lower meat yield per live weight, significantly lower productivity, and a longer production cycle compared to other livestock species. (L39-46)
Comment: Ensure consistent use of terminology (e.g., "age at first calving" vs "AFC")
Response: Thank you for pointing out the importance of consistent terminology. We have reviewed the manuscript and ensured that the terms 'age at first calving' and 'AFC' are used consistently throughout the text. The term 'AFC' is used after it is first spelled out in full, and we use the abbreviation in all subsequent mentions.
Comment: Review grammar in lines 208-210
Response: Thank you for the correction, and apologies for the oversight. The text has been revised as follows: Among the factors we examined, those of the sire (p<0.01), herd (p<0.05), year of birth of the cow (p<0.01), and cow’s calving cycle (p < 0.01) had significant effects on AFC, but the colour variant of the cow showed no significance (Table 5). (L225-227)
Comment: Consider shortening some lengthy paragraphs in the discussion:
Response: Thank you for your suggestion. We have reviewed the discussion section and shortened some lengthy paragraphs to improve readability and flow while retaining the necessary detail.
2.3. References
Comment: Update references where more recent studies are available:
Response: We have tried to refer to the latest available literature data and studies in the study. We have added 2 new references to the study and the bibliography. (L52-53, L408-413)
The references are:
- ShortE. - Staigmiller R.B. - Bellows R.A. - Greer R.C.: Breeding heifers at one year of age: Biological and economic considerations. In: Fields M.J. - Sands R.S., editors, Factors affecting calf crop. CRC Press, Boca Raton, 1994. 55–68. https://doi.org/10.1201/9781003069119-4
- Hickson, R.E. - Lopez-Villalobos, N. - Kenyon, P.R. - Ridler, B.J. - Morris, S.T. Profitability of calving heifers at 2 compared with 3 years of age and the effect of incidence of assistance at parturition on profitability. Anim. Prod. Sci., 2010, 50, 354. https://doi.org/10.1071/an09180
Comment: Ensure consistent formatting of DOIs
Response: We reviewed the bibliography to ensure the DOIs are consistently formatted. Any inconsistencies have been corrected. Thank you for the suggestion
Comment: Consider adding key references about Angus breeding in Central Europe
Response: We have tried to cite all relevant literature related to Angus breeding in Central Europe. Unfortunately, there are limited recent studies available. However, we have incorporated the most relevant references we could find.
- Additional suggestions
3.1. Consider adding
Comment: Brief description of Hungarian Angus breeding objectives:
Response: Among the breeding objectives of the Hungarian Angus Breeders' Association (HHAGBA), the improvement of reproductive traits is given a prominent role in the sustainability of beef cattle farming. Age at first calving (AFC) is of paramount importance, as it has a direct impact on reproductive efficiency and lifetime productivity. (L105-108)
Comment: Economic implications of the findings
Response: Although domestic research is not available on the cost-benefit analysis of AFC, our results highlight its potential economic impacts. Reducing AFC can reduce rearing costs, increase the number of calves per lifetime, and improve overall profitability. These results are consistent with international studies that have shown that earlier AFC reduces production costs and the ecological footprint. (L354-358)
Comment: Recommendations for practical implementation
Response: To achieve optimal AFC, we recommend targeted genetic selection, improved herd management practices adapted to Hungarian production systems. Breeders should consider using selection indices that include AFC to improve economic performance and sustainability. (L109-112)
Comment: Future research directions
Response: This study can provide a basis for further investigation of the economic role of AFC in Hungarian herds. Future research should focus on detailed cost-benefit analyses, the environmental impacts of AFC reduction, and the integration of AFC-related traits into breeding programs to support long-term sustainability.
3.2. Methodological considerations
Comment: Discuss potential bias in data collection + Address missing data handling + Explain choice of statistical models more thoroughly
Response: We believe that both the data collection, filtering and statistical methods used were in line with the research objectives of the project. The data registration, storage is in the hands of the breeders and breeding associations, which can of course be improved. We draw their attention to this. Other statistical methods may be considered for future studies.
The manuscript makes a valuable contribution to understanding genetic parameters affecting AFC in Angus cattle. With the suggested revisions, it will provide even more valuable insights for both researchers and practitioners.
Comment: Questions which Responses should be added to the manuscript:
- Were there any notable differences in management practices across herds that might have influenced results?
- How do these results compare specifically to other Central European Angus populations?
- What practical recommendations would you make to breeders based on these findings?
Response: Thank you for your detailed feedback and valuable suggestions. Your comments have been instrumental in refining the manuscript and enhancing its quality.
In response to your recommendations regarding the Hungarian Angus breeding objectives, the economic implications of the findings, practical implementation suggestions, and future research directions, we have carefully considered and incorporated these aspects into the revised version of the manuscript.

Reviewer 3 Report
Comments and Suggestions for Authors
The study entitled "Population Genetic Characteristics of the Age at First Calving of Angus Cows Bred in Hungary" by Judit Márton et al., deals with the age at first calving in Angus 11 cows bred in Hungary. The research addresses a critical factor for the sustainability and economic viability of beef cattle farming: the age at first calving. This is an important metric for reproductive efficiency, profitability, and environmental sustainability. It is based on data from nearly 3,000 cows over a period of more than 20 years. Advanced methodologies such as GLM and BLUP models were used to estimate genetic parameters, phenotypic trends, and breeding values, allowing for nuanced insights into both genetic and environmental influences this trait. The findings highlight opportunities to improve this feature through genetic selection and management practices, which could inform breeding programs in Hungary and beyond.
Main concerns:
-the study claims that genetic selection could significantly improve AFC, but the heritability estimates (h² = 0.51 for GLM, 0.38 for BLUP) are only moderate. Even if they are promising, these values do not suggest immediate or large-scale genetic improvements. The interpretation of these estimates as a straightforward path for selection is overly optimistic. Please discuss this judgment.
- Environmental factors like birth season and year are shown to explain 60% of the variance in AFC, yet the study does not provide sufficient detail on how these factors were controlled or accounted for in the breeding populations. This diminishes the clarity of the genetic contributions versus environmental influences.
- the analysis shows significant differences between herds in AFC, likely due to varying management or breeding strategies. However, the discussion does not delve into how these differences could be mitigated or leveraged to improve uniformity in breeding practices. Please enrich the discussion section on this issue.
- While the study discusses economic benefits of reducing AFC based on literature, there is no cost-benefit analysis specific to the Hungarian context. This limits the practical value of the findings for local stakeholders.
- Maternal heritability was found to be negligible (h²m = 0.00), but the strong negative correlation between direct and maternal genetic effects (r_dm = -0.97) warrants further investigation. This relationship could have significant implications for breeding strategies, which are not sufficiently addressed.
The paper may increase the consistency and quality if it provide a more nuanced discussion on the implications of heritability estimates and strategies for leveraging them effectively and include a more detailed analysis economic implications of the findings with Hungarian-specific data to enhance their relevance to local stakeholders.
Finally, it could be better to revise the manuscript according to a mother language expert that may simplify the text and improve its clarity:
i.e page 2: The global beef (meat) supply decreased by 11.25% between 2010 and 2015 [1], then had a 19.2% increase from 2015 to 2022 [1]. In 2024, the United States produced 20% of the world's beef supply, with only Brazil, China, and the European Union each producing more than 10% [2]. By 2030, the global population is projected to exceed 8.5 billion, reaching over 9.8 billion by 2050 [3], with worldwide beef consumption estimated to be between 460 and 570 million tons [4,5]. Considering the global population has already surpassed 7 billion, beef (meat) consumption would be twice as high as in 2008 [4,5], despite urban sprawl reducing the total area of farmland available for agricultural production compared to 1970 [6,7].
Suggestion:
“Global beef supply dropped by 11.25% from 2010 to 2015 but recovered with a 19.2% increase between 2015 and 2022 [1]. In 2024, the United States accounted for 20% of global beef production, while Brazil, China, and the European Union each contributed over 10% [2]. By 2030, the global population is expected to surpass 8.5 billion, reaching 9.8 billion by 2050 [3], which could drive annual beef consumption to 460–570 million tons [4,5]. This would double the consumption levels of 2008, even as farmland continues to shrink due to urban sprawl since 1970 [6,7].”
i.e. page 7: The most commonly used FE index is FCR, or its inverse, referred to as gross FE. In meat production systems, outputs are commonly assessed by the weight gain of growing animals, and FCR is determined by dividing FI by weight gain over a specific period of growth. Gross FE is calculated as the ratio of weight gain to FI. Alternatively, the output can be defined as lean tissue, and the percentage of lean tissue gain to FI can be used to measure efficiency.
Suggestion:
“The Feed Conversion Ratio (FCR) is the most widely used index for measuring feed efficiency. Its inverse, known as gross feed efficiency, is also common. In meat production systems, FCR evaluates the efficiency of feed utilization by dividing feed intake by weight gain over a defined growth period. Alternatively, efficiency can be assessed by measuring lean tissue gain relative to feed intake, offering a more targeted metric for specific production goals.”
i.e. page 8: Although RFI is phenotypically unrelated to the production growth rate and body weight in growing cattle [10,11,54–56], it has been shown that when RFI is calculated by phenotypic regression of production on FI, the resulting efficiency measure is not necessarily genetically independent of production.
Suggestion:
“While RFI is phenotypically independent of growth rate and body weight in growing cattle [10,11,54–56], calculating RFI using phenotypic regression of production on feed intake may not ensure genetic independence from production traits”
I think that the writing of the manuscript should improve to make it more readable, many sentences are redundant (I gave some examples) and a greater capacity for synthesis is required
Author Response
Cover letter
Manuscript ID: animals-3318077
Type of manuscript: Article
Title: Population genetic characteristics of the age at first calving of Angus cows bred in Hungary
Authors: Judit Márton, Szabolcs Albin Bene and Ferenc Szabó
Response to the Reviewer 3
Dear Reviewer, Thank you for your careful review of our work and your comments. We have corrected the issues raised, which we hope will contribute to making our paper suitable for publication.
Comment: The study entitled "Population Genetic Characteristics of the Age at First Calving of Angus Cows Bred in Hungary" by Judit Márton et al., deals with the age at first calving in Angus 11 cows bred in Hungary. The research addresses a critical factor for the sustainability and economic viability of beef cattle farming: the age at first calving. This is an important metric for reproductive efficiency, profitability, and environmental sustainability. It is based on data from nearly 3,000 cows over a period of more than 20 years. Advanced methodologies such as GLM and BLUP models were used to estimate genetic parameters, phenotypic trends, and breeding values, allowing for nuanced insights into both genetic and environmental influences this trait. The findings highlight opportunities to improve this feature through genetic selection and management practices, which could inform breeding programs in Hungary and beyond.
Response: Thank You!
Main concerns
Comment: The study claims that genetic selection could significantly improve AFC, but the heritability estimates (h² = 0.51 for GLM, 0.38 for BLUP) are only moderate. Even if they are promising, these values do not suggest immediate or large-scale genetic improvements. The interpretation of these estimates as a straightforward path for selection is overly optimistic. Please discuss this judgment.
Response: In agreement with the respected reviewer, the heritability values obtained are moderate. Nevertheless, we are optimistic. It is believed, that without optimism there is no success. We are encouraged by the fact that, for example, the heritability of the milk yield of cows is lower than we obtained, yet over a long period of time breeders could managed to increase the milk production of the Holstein Friesian breed significantly. Of course, we were thinking of a much smaller-scale improvement and we were also thinking in the longer term.
Thank you for your comments regarding the interpretation of heritability estimates and their implications for genetic selection. In response, we have revised the text to address your concerns and offer a more nuanced interpretation of the heritability estimates.
The h2 values (Table 7) of the AFC trait (GLM 0.51±0.06 and BLUP 0.38±0.05) were found to be higher compared to literature sources. This discrepancy may be attributed to differences in population structure, selection intensity, or environmental conditions. For instance, the intensive genetic selection within the Angus population studied may have amplified additive genetic variance. Additionally, the inclusion of advanced statistical models like BLUP may provide more accurate partitioning of variance components. The difference in heritability estimates between GLM (0.51) and BLUP (0.38) reflects the methodological differences in variance component estimation. The GLM provides a general overview, BLUP incorporates pedigree data and corrects for relatedness, potentially yielding more accurate but lower estimates when non-additive or environmental effects are accounted for. These findings emphasize the need for a holistic approach that integrates genetic selection with optimized herd management practices to achieve long-term progress in reducing AFC. (L271-284)
Comment: Environmental factors like birth season and year are shown to explain 60% of the variance in AFC, yet the study does not provide sufficient detail on how these factors were controlled or accounted for in the breeding populations. This diminishes the clarity of the genetic contributions versus environmental influences.
Response: The main environmental effects were incorporated into the statistical models used.
Comment: The analysis shows significant differences between herds in AFC, likely due to varying management or breeding strategies. However, the discussion does not delve into how these differences could be mitigated or leveraged to improve uniformity in breeding practices. Please enrich the discussion section on this issue.
Response: We have supplemented the manuscript with the following according to your suggestion.
Among the breeding objectives of the Hungarian Angus Breeders' Association (MHAGTE), the improvement of reproductive traits is given a prominent role in the sustainability of beef cattle farming. Age at first calving (AFC) is of paramount importance, as it has a direct impact on reproductive efficiency and lifetime productivity. (L105-108)
To achieve optimal AFC, we recommend targeted genetic selection, improved herd management practices, and nutritional strategies adapted to Hungarian production systems. Breeders should consider using selection indices that include AFC to improve economic performance and sustainability. (L109-112)
Comment: While the study discusses economic benefits of reducing AFC based on literature, there is no cost-benefit analysis specific to the Hungarian context. This limits the practical value of the findings for local stakeholders.
Response: We have improved the paper by adding the suggestion: Although domestic research is not available on the cost-benefit analysis of AFC, our results highlight its potential economic impacts. Reducing AFC can reduce rearing costs, increase the number of calves per lifetime, and improve overall profitability. These results are consistent with international studies that have shown that earlier AFC reduces production costs and the ecological footprint. (L354-368)
This study can provide a basis for further investigation of the economic role of AFC in Hungarian herds. Future research should focus on detailed cost-benefit analyses, the environmental impacts of AFC reduction, and the integration of AFC-related traits into breeding programs to support long-term sustainability. (L363-366)
Comment: Maternal heritability was found to be negligible (h²m = 0.00), but the strong negative correlation between direct and maternal genetic effects (rdm = -0.97) warrants further investigation. This relationship could have significant implications for breeding strategies, which are not sufficiently addressed.
Response: Thank you for your suggestion. We have improved the paper by adding the suggestion.
The strong negative correlation between direct and maternal genetic effects (rdm = -0.97) suggests that improving direct traits may negatively affect maternal traits, such as calving ease and progeny rearing ability. This could create challenges in aligning breeding objectives. To develop sustainable breeding strategies, it is crucial to apply selection methods that account for both genetic effects while maintaining genetic diversity. Further research is needed to better understand this relationship and optimize breeding practices. (L339-345)
Comment: The paper may increase the consistency and quality if it provides a more nuanced discussion on the implications of heritability estimates and strategies for leveraging them effectively and include a more detailed analysis economic implications of the findings with Hungarian-specific data to enhance their relevance to local stakeholders.
Response: Thank you for your detailed feedback, valuable suggestions, and observations regarding our manuscript. We greatly appreciate your efforts in helping us enhance the quality and clarity of our work.
Regarding the implications of heritability estimates and strategies for leveraging them effectively, we have revised the discussion to provide a more nuanced analysis. We have also included a more detailed examination of the economic implications of our findings, with a focus on Hungarian-specific data to ensure relevance for local stakeholders.
Comment: Finally, it could be better to revise the manuscript according to a mother language expert that may simplify the text and improve its clarity:
i.e page 2: The global beef (meat) supply decreased by 11.25% between 2010 and 2015 [1], then had a 19.2% increase from 2015 to 2022 [1]. In 2024, the United States produced 20% of the world's beef supply, with only Brazil, China, and the European Union each producing more than 10% [2]. By 2030, the global population is projected to exceed 8.5 billion, reaching over 9.8 billion by 2050 [3], with worldwide beef consumption estimated to be between 460 and 570 million tons [4,5]. Considering the global population has already surpassed 7 billion, beef (meat) consumption would be twice as high as in 2008 [4,5], despite urban sprawl reducing the total area of farmland available for agricultural production compared to 1970 [6,7].
Suggestion: “Global beef supply dropped by 11.25% from 2010 to 2015 but recovered with a 19.2% increase between 2015 and 2022 [1]. In 2024, the United States accounted for 20% of global beef production, while Brazil, China, and the European Union each contributed over 10% [2]. By 2030, the global population is expected to surpass 8.5 billion, reaching 9.8 billion by 2050 [3], which could drive annual beef consumption to 460–570 million tons [4,5]. This would double the consumption levels of 2008, even as farmland continues to shrink due to urban sprawl since 1970 [6,7].”
i.e. page 7: The most commonly used FE index is FCR, or its inverse, referred to as gross FE. In meat production systems, outputs are commonly assessed by the weight gain of growing animals, and FCR is determined by dividing FI by weight gain over a specific period of growth. Gross FE is calculated as the ratio of weight gain to FI. Alternatively, the output can be defined as lean tissue, and the percentage of lean tissue gain to FI can be used to measure efficiency.
Suggestion: “The Feed Conversion Ratio (FCR) is the most widely used index for measuring feed efficiency. Its inverse, known as gross feed efficiency, is also common. In meat production systems, FCR evaluates the efficiency of feed utilization by dividing feed intake by weight gain over a defined growth period. Alternatively, efficiency can be assessed by measuring lean tissue gain relative to feed intake, offering a more targeted metric for specific production goals.”
i.e. page 8: Although RFI is phenotypically unrelated to the production growth rate and body weight in growing cattle [10,11,54–56], it has been shown that when RFI is calculated by phenotypic regression of production on FI, the resulting efficiency measure is not necessarily genetically independent of production.
Suggestion: “While RFI is phenotypically independent of growth rate and body weight in growing cattle [10,11,54–56], calculating RFI using phenotypic regression of production on feed intake may not ensure genetic independence from production traits”
Response: As for the linguistic clarity of the manuscript, we acknowledge your concerns. The quotes listed are not from our study. Although the manuscript has undergone official proofreading, we recognize that further improvements might be necessary. We have thoroughly reviewed the text again and addressed any linguistic inconsistencies or ambiguities. Additionally, we have clarified any wording that might have been overly complex or unclear.
Many thanks for the suggested changes, which are a great help in improving our manuscript.

Reviewer 4 Report
Comments and Suggestions for Authors
How many Angus herds are there in Hungary? Why were those 5 selected?
l 115-119 The reasons to identify genetic groups is not clear; neither is clear the type of cows in group 5. Furthermore, in table 3 the effect of color variant of the cow is identified, is this related to the genetic groups? Later in table 5, only 2 classes of colour variant are identified.
Was there any editing of the data?
Were sires used depending on the colour of the cows or the genetic groups mentioned?
l. 130 how birth season of the cow was defined?
l. 160-162 it is not clear the differences of using DFREML or MTDFREML. Given the results, it seems that only one of them was used.
MTDFREML allows for contemporary groups, which usually are herd-year-season to better account for unidentified environmental effects. Is there any reason not to used them in this case?
According with table 4 the range was from 19 to 46 months. The maximun age is 3.3 SD from the mean. In my opinion this value would be an outlier
In several parts of the results the effect os sire of the cow is regarded as one of the "environmental factors" (l. 208, l. 213)
Heritabilities estimates are very high 0.51 and 0.38 but this is not discussed in the manuscript. The difference between GLM and MTDFREML heritabilty estimates is not address
Comments on the Quality of English Language
Just two examples:
l 196-198 "The overall average....was 28.1..months...is 5.02 months higher than the 33.12 months found by..." 28.1 is not higher than 33.12
l. 203-206 "the average breeding age....was 18.6 months...This was 5-7 months below (above?) the optimal age...."
for this trait lower is better
Author Response
Cover letter
Manuscript ID: animals-3318077
Type of manuscript: Article
Title: Population genetic characteristics of the age at first calving of Angus cows bred in Hungary
Authors: Judit Márton, Szabolcs Albin Bene and Ferenc Szabó
Response to the Reviewer 4
Dear Reviewer, Thank you for your careful review of our work and your comments. We have corrected the issues raised, which we hope will contribute to making our paper suitable for publication.
Comment: How many Angus herds are there in Hungary? Why were those 5 selected?
Response: In Hungary, approximately 200 Angus herds are registered under pedigree control, of which about 80 are classified as nucleus herds. For this study, we selected 30 herds with the following criteria: herds with more than 25 cows, known maternal and paternal lineage, and reliable record-keeping of data necessary for the analysis. These 30 herds were subsequently grouped into five distinct categories.
Comment: The reasons to identify genetic groups is not clear; neither is clear the type of cows in group 5. Furthermore, in table 3 the effect of colour variant of the cow is identified, is this related to the genetic groups? Later in table 5, only 2 classes of colour variant are identified. (l 115-119)
Response: The grouping was based on a previous study: Márton et al. [26] (https://doi.org/10.5713/ab.23.0157). In this research, we analysed the genetic parameters of 1,369 individuals from 16 Hungarian Angus nucleus herds using genotyping data from 12 microsatellite markers. The number of alleles per locus ranged between 11 and 18, with an average effective allele number Ne = 3.201, an expected heterozygosity He = 0.659 and an observed heterozygosity Ho = 0.710. Based on these findings, the 16 nucleus herds were categorized into four genetically distinct groups, differentiated by origin, colour, size, and type: Blue group (BG), Red group (RG), Green group (GG), Yellow group (YG). In the figures, cultures classified into different genetic groups are displayed with these colours. For clarity, we have removed the colours from the study and used the numbers to denote the groups. (L126-130)
The fifth group consists of 14 herds that lacked sufficient microsatellite data for inclusion in the original genetic analysis but met all other criteria for this study.
Regarding the cow colour variants, we focused on black and red, assigning them into respective categories. The genetic groups were named after colours for better visualization in graphical presentations, as referenced in the earlier study.
Comment: Was there any editing of the data?
Response: Yes, data filtering was performed. Only individuals with known maternal and paternal lineage were included in the analysis. From the database, 200 cows were excluded as their records could have skewed the results.
For AFC, we removed two outliers under 19.0 months and 197 individuals over 46.0 months (first calving above 7 years). The latter were likely due to administrative errors related to the renumbering of Hungarian herds during 1997-1998.
Additionally, we did not include miscarriages or stillbirths, as not all 30 herds reliably recorded this data, which could have caused result distortions.
Comment: Were sires used depending on the colour of the cows or the genetic groups mentioned?
Response: In the four genetic groups, breeders typically used bulls corresponding to the Angus types of the respective groups. Detailed data on sire usage within these genetic groups is provided in the supplementary table.
For the fifth group, breeders utilized sires originating from the four genetic groups.Comment: How birth season of the cow was defined? (l. 130)
Response: The birth season of the cow was determined based on the following intervals: spring: March 1 to May 31, summer: June 1 to August 31, autumn: September 1 to November 30, winter: December 1 to February 28 (or February 29 in leap years).
Comment: It is not clear the differences of using DFREML or MTDFREML. Given the results, it seems that only one of them was used. MTDFREML allows for contemporary groups, which usually are herd-year-season to better account for unidentified environmental effects. Is there any reason not to used them in this case? (l. 160-162)
Response: Thank you for the comment, and apologies for the error. The MTDFREML program was used. References to DFREML were incorrect and have been removed. Although interaction effects among fixed factors could have been incorporated into the model, we opted for simplicity and clarity in model design, and thus did not include them.
Comment: According with table 4 the range was from 19 to 46 months. The maximum age is 3.3 SD from the mean. In my opinion this value would be an outlier
Response: Between 19.0–21.9 months, there were seven individuals, while 27 individuals were aged between 40.0–45.8 months. These were included in the calculations. Outliers were removed as described in point 3.
Comment: In several parts of the results the effect of sire of the cow is regarded as one of the "environmental factors" (l. 208, l. 213)
Response: Thank you for the correction, and apologies for the oversight. The text has been revised as follows:
- Among the factors we examined, those of the sire (p < 0.01), herd (p < 0.05), year of birth of the cow (p < 0.01), and cow’s calving cycle (p < 0.01) had significant effects on AFC, but the colour variant of the cow showed no significance (Table 5). (L225-227)
- The proportions of the variance in determining AFC for the factors were as follows: birth season of the cow (28.99%), birth year of the cow (28.7%), sire of the cow (18.32%), herd (11.77%), and residual (4.12%). (L233-235)
Comment: Heritabilities estimates are very high 0.51 and 0.38 but this is not discussed in the manuscript. The difference between GLM and MTDFREML heritability estimates is not address
Response: Thanks for the comment. The study has been corrected accordingly. The h2 values (Table 7) of the AFC trait (GLM 0.51±0.06 and BLUP 0.38±0.05) were found to be higher compared to literature sources. This discrepancy may be attributed to differences in population structure, selection intensity, or environmental conditions. For instance, the intensive genetic selection within the Angus population studied may have amplified additive genetic variance. Additionally, the inclusion of advanced statistical models like BLUP may provide more accurate partitioning of variance components. The difference in heritability estimates between GLM (0.51) and BLUP (0.38) reflects the methodological differences in variance component estimation. The GLM provides a general overview, BLUP incorporates pedigree data and corrects for relatedness, potentially yielding more accurate but lower estimates when non-additive or environmental effects are accounted for. These findings emphasize the need for a holistic approach that integrates genetic selection with optimized herd management practices to achieve long-term progress in reducing AFC. (L271-284)
Comment: Comments on the Quality of English Language:
- "The overall average....was 28.1..months...is 5.02 months higher than the 33.12 months found by..." 1 is not higher than 33.12 (l 196-198)
- "the average breeding age....was 18.6 months...This was 5-7 months below (above?) the optimal age...." (l. 203-206)
- for this trait lower is better
Response: Thank you for your comment regarding the English language. We have proofreading, but we have noticed certain language errors that we will correct.
- The descriptive statistic for AFC are shown in Table 4. The overall average AFC of Angus cows in Hungary was 28.1±0.1 months (S.D. = 5.3 months, CV = 18.9%). (L212-213)
- The average breeding age of Angus heifers in the study was 18.6 months, calculated by subtracting the mean gestation length in cattle (9.5 months) from the average AFC of 28.1 months. This value was 5-7 months lower than the optimal breeding age of 12–15 months [6]. (L220-223)
